# The Role of NLRP3 Inflammasome in IgA Nephropathy

**DOI:** 10.3390/medicina59010082

**Published:** 2022-12-30

**Authors:** Xiaofang Wu, Lei Zhao, Kailong Li, Jurong Yang

**Affiliations:** Department of Nephrology, The Third Affiliated Hospital of Chongqing Medical University, Chongqing 400120, China

**Keywords:** autophagy, exosomes, IgA nephropathy, NF-κB, NLRP3 inflammasome, ROS

## Abstract

Immunoglobulin A nephropathy (IgAN) is the most common primary glomerular disease worldwide today. The NLRP3 inflammasome is a polyprotein complex and an important participant in inflammation. Accumulating studies have shown that the NLRP3 inflammasome participates in a variety of kidney diseases, including IgAN. This review focuses on the role of the NLRP3 inflammasome in IgAN and summarizes multiple involved pathways, which may provide novel treatments for IgAN treatment.

## 1. Introduction

Immunoglobulin A nephropathy (IgAN) is the most common variety of primary glomerular disease worldwide today, and the deposition of IgA immune complexes (IgA-ICs) within glomeruli is the most outstanding characteristic [1,2,3]. The deposition of immune complexes can activate mesangial cell proliferation and induce cytokine secretion, resulting in inflammation and ultimately leading to kidney damage [3,4]. Studies have shown that approximately one third of IgAN patients progress to end-stage renal disease (ESRD) within 20 years [1,5].

The nucleotide-binding oligomerization domain (NOD)-like receptors (NLRs) compose a group of pattern recognition receptors (PRRs) and participate in inducing host innate immune responses to cellular injury [6]. The NLR family pyrin domain-containing 3 (NLRP3) is one of the best understood members and the core protein of the NLRP3 inflammasome [6,7]. The NLRP3 inflammasome is an approximately 700 kD polyprotein complex and an important participant in inflammation, which consists of NLRP3, apoptosis-associated speck-like protein containing a caspase recruitment domain (ASC) and the protease caspase-1 [6,7,8]. Active caspase-1 cleaves the cytokines pro-interleukin-1β (pro-IL-1β) and pro-interleukin-18 (pro-IL-18) into their mature and biologically active forms IL-1β and IL-18, inducing inflammation and tissue damage [9].

NLRP3 inflammasome activation is a two-step process, consisting of priming and activation. A priming signal is required for its activation, such as ligands for Toll-like receptors (TLRs), NLRs or cytokine receptors, which trigger the transcription of nuclear factor-kappa B (NF-κB) [9,10]. NF-κB promotes the expression of NLRP3 and pro-IL-1β, but does not upregulate pro-IL-18, ASC or pro-caspase-1 [9,11]. Inflammasome can be activated via both exogenous pathogen-associated molecular patterns (PAMPs) and endogenous damage-associated molecular patterns (DAMPs) [10]. It happens when exposed to stimulus such as reactive oxygen species (ROS), mitochondrial dysfunction, lysosomal damage, ionic flux, pathogen-associated RNA and bacterial or fungal toxins [9,10,12]. NLRP3 inflammasome activation happens not only in immune cells, such as macrophages and dendritic cells, but also in kidney cells, such as podocytes, mesangial cells, renal tubular epithelium, etc. [7,8,13].

Based on the above findings, accumulating studies have shown that the NLRP3 inflammasome participates in a variety of kidney diseases, including diabetic nephropathy (DN), obesity-related kidney disease, acute kidney injury (AKI), crystal-related nephropathy, lupus nephritis (LN) and IgAN [7,8,9,10,12,14,15]. Previous studies have demonstrated that IgA-ICs can initiate the activation of the NLRP3 inflammasome in IgAN macrophages and podocytes [13,16]. Additionally, the markers of NLRP3 inflammasome activation, IL-18 and IL-1β, were elevated in IgAN patients [16,17,18]. Studies in IgAN mouse models have indicated that NLRP3 inflammasome-related pathways may be strongly associated with the progression of IgAN [17,19]. Another study found that colorectal neoplasia differentially expressed (CRNDE) exacerbates IgAN progression by promoting NLRP3 inflammasome activation in macrophages, and the inhibition of CRNDE promoted NLRP3 degradation [20]. These studies revealed that the inhibition of the NLRP3 inflammasome may be a beneficial strategy for the treatment of IgAN. Therefore, this review focuses on the role of the NLRP3 inflammasome in IgAN and identifying novel treatments for IgAN patients.

## 2. The NLRP3 Inflammasome and Related Pathways

### 2.1. The NLRP3 Inflammasome and NF-κB Pathway

Previous studies have demonstrated that NF-κB plays a pivotal role in the pathogenesis of inflammation, and NF-κB expression is correlated with the poor prognosis of IgAN patients [21,22]. Varieties of endogenous or exogenous stimuli could trigger the transcription of NF-κB, which is the main signal inducing the activation of the NLRP3 inflammasome [10,23].

Studies have illustrated that activation of the NF-κB/NLRP3 pathway might participate in the pathogenesis of inflammation in IgAN, and inhibiting NLRP3 activation can alleviate the inflammation [4,17,19,24,25,26,27]. For example, He L. et al. found that triptolide could down-regulate serum levels of IL-1β and IL-18 and may exert an anti-inflammatory effect by suppressing NLRP3 and TLR4 expression on IgAN rats [27]. Another study on rats found that artemisinin and hydroxychloroquine combination therapy exert protective effects on IgAN by inhibiting NF-κB signaling and NLRP3 inflammasome activation [25]. A recent study also discovered that IgAN mice benefited from compound K (a major absorbable intestinal bacterial metabolite of ginsenosides) and Icariin (a major constituent of flavonoids isolated from plants of the genus Epimedium) by inhibiting the NF-κB/NLRP3 pathway, respectively [17,26].

In summary, these findings indicate that the NF-κB/NLRP3 pathway is essential in the pathogenesis of IgAN, and the inhibition of its activation may be an effective therapeutic method.

### 2.2. The NLRP3 Inflammasome and Autophagy

Autophagy, a vital intracellular process that degrades dysfunctional proteins and organelles (e.g., mitochondria) via lysosome-mediated degradation, clears damaged intracellular pathogens and regulates the diverse immune system such as antigen presentation [28,29,30]. Autophagy has now been identified as an important regulator of the NLRP3 inflammasome [30,31,32,33]. Previous studies have shown that inflammatory signals lead to an induction of autophagy, which plays a negative role in the activation of the NLRP3 inflammasome and promotes cell survival and restores tissue homeostasis after damage in autoimmune diseases, including IgAN [16,26,28].

Accumulating evidence has indicated that the regulation of inflammasomes and autophagy may be the key for the treatment of multiple diseases, including kidney disease [30,31,32,33]. Qu et al. showed that cisplatin may induce kidney injury by inhibiting autophagy and activating NLRP3 inflammasomes [34]. Additionally, in their later study, they found that astragaloside IV could alleviate cisplatin-induced AKI by inducing autophagy and limiting the expression of the NLRP3 inflammasome [35]. Recent reviews also outlined that autophagy inhibits inflammatory responses induced in AKI through the inhibition of inflammasome activation, suggesting that the enhancement of autophagy, such as the use of autophagy activators, might be a potential target for the treatment of AKI [33,36].

The relation between NLRP3 and autophagy also plays a vital role in the development of IgAN. In mouse models of progressive IgAN, researchers showed that resveratrol inhibits the NLRP3 inflammasome activation by augmenting autophagy and preserving mitochondrial integrity [37]. Additionally, in cultured macrophages, Tris dibenzylideneacetone dipalladium (Tris DBA), a small-molecule palladium complex, was found to inhibit the activation of the NLRP3 inflammasome and regulate the autophagy/NLRP3 inflammasome axis through SIRT1 and SIRT3 [16]. In addition, a recent study in Taiwan found that compound K inhibited the activation of the renal NLRP3 inflammasome in treated IgAN mice, and increased induction of autophagy in IgA-IC-primed macrophages, revealing the protective mechanisms of autophagy in IgAN [26]. In their later study in vitro and vivo, LCC18, a benzamide-linked small molecule, was found to improve renal function and reduce proteinuria in IgAN by blocking the priming of the NLRP3 inflammasome and inhibiting its activation through autophagy induction, further confirming the positive effect of autophagy in IgAN [38].

Collectively, these results suggested that inhibiting NLRP3 activation through autophagy induction may be a potential novel therapeutic approach for IgAN.

### 2.3. The NLRP3 Inflammasome and Mitochondrial Reactive Oxygen Species

Previous studies have indicated that the most typical mechanism for activating the NLRP3 inflammasome is the production of ROS, especially mitochondrial ROS (mtROS) [10,39,40,41]. Mitochondrial dysfunction has long been considered a necessary factor in triggering NLRP3-mediated inflammation, and overproduction of mtROS is a key factor in NLRP3 inflammasome activation [39,41]. Excessive mtROS production induces thioredoxin (TRX) separation from thioredoxin-interacting protein (TXNIP), and then the latter binds to NLRP3 and activates the NLRP3 inflammasome [39,42].

A growing number of studies have revealed the role of blocking mtROS in kidney diseases, such as ischemic and cisplatin-induced AKI, DN, etc. [39,43,44,45,46,47]. A previous study found that Mito TEMPO, a mitochondria-targeted antioxidant, can inhibit mtROS overproduction and NLRP3 inflammasome activation, and it verified that the NLRP3 inflammasome can be activated via the mROS-TXNIP-NLRP3 signal pathway, providing a potential therapeutic target for ischemic AKI [43]. Han et al. also found that oral administration of the mitochondria-targeted antioxidant MitoQ reduced mtROS levels, thereby inhibiting the TXNIP/NLRP3/IL-1β signaling pathway, leading to the alleviation of kidney injury in DN mice [39].

The ROS signaling pathway has also been shown to be involved in IgAN [19,48]. It has been well-recognized that albuminuria is a risk factor of IgAN, and albuminuria triggers mitochondrial dysfunction and mtROS generation, resulting in renal tubular inflammation through mtROS-meditated activation of the NLRP3 inflammasome [24,49]. A previous study found that IgA ICs could induce the activation of the NLRP3 inflammasome through ROS in macrophages [48]. Yang et al. found in induced accelerated progressive IgAN mice that antroquinonol (a pure active compound from Antrodia camphorata mycelium) promoted the Nrf2 antioxidant pathway, inhibited NLRP3 inflammasome activation and significantly improved renal function [50]. Additionally, in IgA-IC-primed macrophages, they discovered that antroquinonol inhibited NLRP3 inflammasome activation by reducing ROS production [50]. Hua et al. also found that osthole inhibited ROS production, activation of NF-κB and the NLRP3 inflammasome, exerting its reno-protective effects on the progression of IgAN both in vitro and in vivo [19]. Based on these findings, ROS inhibition may be a potential choice to inhibit NLRP3 activation and reduce inflammation in IgAN.

### 2.4. The NLRP3 Inflammasome and Exosomes

Exosomes are small extracellular vesicles (30–150 nm) secreted by all healthy and abnormal cells and are abundant in all bodily fluids [51,52]. Exosomes contain specific protein, lipid, RNA and DNA compositions that are derived from the endocytosis membrane and can transmit signals to recipient cells, playing a key role in intercellular communications [51,53,54]. Exosomes play significant roles in inflammation and immune response, and they are considered promising biomarkers for diagnosis and therapy in various diseases, including kidney diseases such as LN, AKI, DN and IgAN [51,55,56,57,58,59,60].

Emerging evidence has revealed the relationship between exosomes and the NLRP3 inflammasome [61,62,63,64]. Recent studies have shown that exosomes can influence the course of NLRP3 inflammasome-associated diseases by secreting different substances that affect key molecules in the canonical pathway [61,62]. Dai et al. discovered that exosomes relieve myocardial ischemia/reperfusion injury by inactivating the TLR4/NF-κB/NLRP3 inflammasome signaling pathway in a neonatal rat model induced by ischemia/reperfusion [63]. In another rat model, Tang et al. found that exosomal miR-320b can directly target NLRP3 and inhibit pyroptosis, thereby protecting the myocardium from ischemia/reperfusion injury by inhibiting pyroptosis [65].

Recent research also focused on the mechanism by which exosomes mediate inflammation in IgAN [4,25]. Bai et al. found that artemisinin and hydroxychloroquine combination therapy could significantly promote the secretion of exosomes in the renal tissue of IgAN rats and inhibit the expressions of NF-κB signal and NLRP3 inflammasome-related protein [25]. Subsequently, Li et al. found that Zhen-wu-tang (a well-known traditional Chinese formula) regulated exosome secretion, which influenced the NF-KB/NLRP3 signaling pathway in the human mesangial cell proliferation model, and it could also reinforce the secretion of exosomes in an IgAN rat model [4]. These results have provided new evidence that enhancing the secretion of exosomes to inhibit the NF-κB/NLRP3 signaling pathway is a promising approach for IgAN treatment.

Moreover, a recent study in IgAN patients found that supplementation of probiotics can significantly improve gut dysbiosis and ameliorate IgAN by inhibiting the NLRP3/ASC/Caspase-1 signaling pathway [66]. A summary of the main publications about treatments related to pathways between IgAN and the NLRP3 inflammasome is shown in Table 1. Figure 1 illustrates the related pathways between IgAN and the NLRP3 inflammasome and the plausible mechanism of treatments.

Other studies have also shown that the renin–angiotensin–aldosterone system (RAAS) and endoplasmic reticulum stress (ERS) can regulate the NLRP3 inflammasome and play an important part in the development of renal diseases, including DN, obesity-related kidney disease and AKI [8,67,68,69,70]. The relationship between the NLRP3 inflammasome and RAAS and ERS is expected to be found in IgAN.

## 3. Conclusions

In this review, we summarized information regarding multiple pathways between IgAN and the NLRP3 inflammasome, including the NF-κB/NLRP3 pathway, autophagy, mtROS production and exosomes. These studies suggest that NLRP3 could be a promising therapeutic target for the design of a novel therapeutic treatment for IgAN. In the future, these pathways need to be completely understood and are worthy of further investigation in humans.

## Figures and Tables

**Figure 1 medicina-59-00082-f001:**
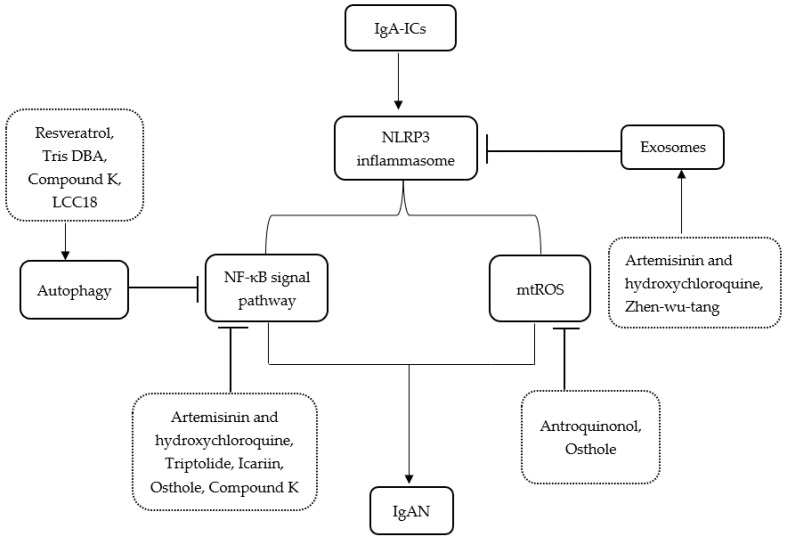
Schematic representation for the related pathways between IgAN and the NLRP3 inflammasome and the plausible mechanism of treatments. IgA-ICs, IgA immune complexes; IgAN, IgA nephropathy; Tris DBA, Tris dibenzylideneacetone dipalladium; ROS, reactive oxygen species.

**Table 1 medicina-59-00082-t001:** Studies about treatments related to pathways between IgAN and NLRP3 inflammasome.

Treatments	Related Pathways	Models	Year
Zhen-wu-tang [4]	Regulates exosomes to inhibit NF-κB/NLRP3 pathway	IgAN rat model and human renal tubular epithelial cells (HK-2)	2020
Tris DBA [16]	Inhibits the activation of NLRP3 inflammasome through SIRT1- and SIRT3-mediated autophagy	IgAN mouse model and cultured macrophage cells	2020
Icariin [17]	Ameliorates IgA nephropathy by inhibition of NF-κB/NlRP3 pathway	IgAN rat model and BMDCs	2016
Osthole [19]	Inhibits ROS generation and NF-κB/NLRP3 pathway	Progressive IgAN mouse model, cultured macrophage cells and mesangial cells	2013
Artemisinin and hydroxychloroquine combination therapy [25]	Suppress NF-κB signaling and NLRP3 inflammasome activation by exosomes	IgAN rat model and human renal tubular epithelial cells (HK-2)	2019
Compound K [26]	Inhibits NF-κB/NLRP3 inflammasome and enhance autophagy and SIRT1	IgAN mouse model	2020
Triptolide [27]	Down-regulates NLRP3 and TLR4 expression	IgAN rat model	2015
Resveratrol [37]	Inhibits NLRP3 inflammasome activation by augmenting autophagy and preserving mitochondrial integrity	IgAN mouse model	2015
LCC18 [38]	Inhibits the MAPKs/COX-2 axis-mediated priming of the NLRP3 inflammasome and inhibits NLRP3 inflammasome activation through autophagy induction	IgAN mouse model and cultured macrophage cells	2021
Probiotics [66]	Improve gut dysbiosis and inhibit the NLRP3/ASC/Caspase-1 signaling pathway	IgAN patients and IgAN mouse model	2022

IgAN, IgA nephropathy; Tris DBA, Tris dibenzylideneacetone dipalladium; BMDCs, macrophages and bone-marrow-derived dendritic cells; ROS, reactive oxygen species; TLR4, Toll-like receptor 4; MAPK, mitogen-activated protein kinases; COX, cyclooxygenase-2; ASC, apoptosis-associated speck-like protein that contains a caspase recruitment domain.

## Data Availability

Not applicable.

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
