# Peer review of "The Role of NLRP3 Inflammasome in IgA Nephropathy"

_medicina, 2022, doi:10.3390/medicina59010082_

Round 1

Reviewer 1 Report

This review article is written in a relatively easy-to-understand manner on the relationship between the NLRP3 inflammasome and potential therapeutic targets. It is recommended to review it for correction of some English uppercase and lowercase letters.

Author Response

 Thank you so much for your advice. It was our careless to mix some uppercase and lowercase letters. We have corrected them in the review,such as Compound K, Astragaloside IV, Zhen-wu-tang,and et al. Thank you again for your approval.

Reviewer 2 Report

The work of Xiaofang Wu, Lei Zhao, Kailong Li, Jurong Yang entitled: "The Role of NLRP3 Inflamansome in IgA Nephropathy" is an interesting work summarizing the role of NLRP3 in IgA nephropathy. The work is interesting and constitutes a short review of the existing literature. The manuscript is prepared and structured correctly, however, its reception could be improved. Here are some of my comments:

1. In the search engines of scientific articles such as: Scopus or pubmed after entering the keywords: NLRP3, Nephropatii IgA. the Scopus database displays 803 articles, while the pubmed database displays 820. Please verify the citations, and present the publication search scheme, with the development of the path of exclusion or inclusion of publications along with a description of keywords, a criterion for preparing the review.

2. Please check the manuscript for minor editorial corrections spaces, periods e.g.: lines 56, 67

3. Please consider adding a picture to make it easier to read the content of the article.

Author Response

Response 1: Thank you for your sincere advice. We again used the search engines of Scopus and pubmed and entered the keywords”NLRP3, IgA”, the pubmed database displays 39 articles, the Scopus database displays 12 articles. After entering the keywords” NLRP3, IgA nephropathy”, or “NLRP3, Nephropatii IgA” the pubmed database displays 21 articles. We have read all of the searched articles and other related articles, such as using the key words ”NLRP3, NF-κB”, ”NLRP3, autophagy”, “NLRP3, ROS”, ”NLRP3, exosomes “, and et al. If we have misundertood your opinion, please don’t hesitate to contact us again. Thanks again.

Response 2: Thank you for your advice. It was our careless to make this mistake. We have corrected it,  and checked all the manuscript and revised it.

Response 3: Thank you so much for this pertinent Suggestions. We add a picture in the manuscript, it does make it easier to read. Thank you again.
